# Evaluation of the Effect of Pb Pollution on Avian Influenza Virus-Specific Antibody Production in Black-Headed Gulls (*Chroicocephalus ridibundus*)

**DOI:** 10.3390/ani13142338

**Published:** 2023-07-18

**Authors:** Nana Ushine, Makoto Ozawa, Shouta M. M. Nakayama, Mayumi Ishizuka, Takuya Kato, Shin-ichi Hayama

**Affiliations:** 1Laboratory of Wildlife Medicine, Department of Veterinary Medicine, Nippon Veterinary and Life Science University, Musashino 180-0023, Japan; tkato@nvlu.ac.jp (T.K.); hayama@nvlu.ac.jp (S.-i.H.); 2Laboratory of Animal Welfare, Department of Animal Health Technology, Yamazaki University of Animal Health Technology, Hachioji 192-0364, Japan; 3Joint Faculty of Veterinary Medicine, Kagoshima University, Kagoshima 890-0065, Japan; mozawa@vet.kagoshima-u.ac.jp; 4School of Veterinary Medicine, The University of Zambia, Lusaka P.O. Box 32379, Zambia; shoutanakayama0219@gmail.com; 5Laboratory of Toxicology, Department of Environmental Veterinary Sciences, Faculty of Veterinary Medicine, Hokkaido University, Sapporo 060-0808, Japan; ishizum@vetmed.hokudai.ac.jp

**Keywords:** antibody production, avian influenza, black-headed gull, Pb pollution, spread of zoonotic diseases

## Abstract

**Simple Summary:**

Lead (Pb) is an environmental pollutant with reported contamination from mammals to fish. In particular, birds are often reported to be poisoned, and populations of some avian species have even reduced. In this study, we examined the causal relationship between the effects on immune function, which is one of the biological effects caused by Pb pollution, and focused on the expression of avian influenza virus (AIV) antibodies. Among black-headed gulls (*Chroicocephalus ridibundus*) that migrate in winter, two populations were targeted in this study: one was overwintering in Tokyo Bay (Tokyo Bay population; TBP) and the other in Mikawa Bay (Mikawa Bay population; MBP). Pb pollution was evaluated based on blood lead levels (BLL) and antibody positive rate (infection history), and antibody titer was evaluated using serum. The results indicated that the antibody titer was significantly decreased owing to increased BLL in MBP. There were no significant year-to-year differences in BLL or antibody titer. In the wintering period, antibody titer was also significantly decreased owing to increased BLL in TBP. The findings of this study indicated that Pb pollution had a possible negative effect on the antibody production of AIV.

**Abstract:**

Lead (Pb), an environmental pollutant, has been widely reported to have contaminated mammals, including humans and birds. This study focuses on the effects of Pb pollution on avian influenza virus (AIV) antibody production. A total of 170 black-headed gulls (*Chroicocephalus ridibundus*) were captured in Tokyo Bay (TBP) from January 2019 to April 2020 and in Mikawa Bay (MBP) from November 2019 to April 2021. The gulls were weighed, subjected to blood sampling, and released with a ring band on their tarsus. The samples were used to measure blood Pb levels (BLL) and AIV-specific antibodies. The BLL were compared using the Wilcoxon two-sample test between the period when black-headed gulls arrived and the wintering period, defined by the number of gulls counted in each area. A significant increase was found in the TBP. A decrease in BLL significantly increased antibody titer during wintering in TBP and MBP. Pb pollution had a negative effect on the production of AIV antibodies. These findings suggest that wild birds that were contaminated by Pb in the environment may facilitate the spread of zoonotic diseases, further increasing the possibility that environmental pollutants may threaten human health.

## 1. Introduction

Avian influenza (AI) is an infectious disease prevalent in wild birds and poultry [1,2]. Recently, high-pathogenicity viruses have been reported to infect humans in East Asia and northern Africa [3,4]. Since 2004, outbreaks of AI in Japan have recurringly occurred during the winter [5,6,7]. Avian influenza viruses (AIV) are considered to be introduced into Japan by transportation associated with economic activities and migratory birds [8]. In countries like Japan, where AI typically occurs in the winter, migratory birds that fly in during the autumn are considered to be the main carriers of the virus [9,10].

Individuals infected with AIV produce antibodies, and when initial infection occurs, the antibody titers peak within seven days post infection (dpi) and persist for several days thereafter [11]. Boltz et al. [12] observed long-term antibody titers in chickens (*Gallus gallus domesticus*) primed with the H5N3 subtype and reported that antibody titers persisted for approximately seven months without significantly decreasing. Based on these studies, it is assumed that the antibody titer of AIV becomes constant at approximately seven dpi and persists slowly for several months if it is the first infection.

Heavy metal Pb is an environmental pollutant with reported contamination effects on mammals and avian species [13]. In terms of wildlife, several studies have reported Pb pollutants in avian species, revealing that some avian species are highly sensitive to Pb [14]. Pb has various effects depending on the pollution levels [15]. Among the symptoms of pollution, sudden death owing to acute poisoning, eggshell malformation, and behavioral abnormalities are feared to affect the species’ survival [16]. For example, the number of California condors (*Gymnogyps californianus*) declined sharply from 1982 to 1986 due to the consumption of meat contaminated with Pb shot, and presently there are only approximately 1000 birds [17]. The white-headed duck (*Oxyura leucocephala*) population has rapidly declined within 10 years and is endangered due to contamination from feeding on Pb shot [17,18].

Pb contamination has an impact on immunity. Although the mechanism of the effect of Pb on immune functions is unclear [19], it affects immunity in mammals and avian species [20,21]. Among immune cells, CD4-positive T cells react sensitively [22]. Pb pollution causes dysfunction of lymphocytes, impaired cytokine synthesis [23], and impaired antibody production [24]. Furthermore, in chickens, Pb contamination significantly reduced Th1/Th2, which may have a particularly inhibitory effect on the production of antiviral antibodies [25]. Exposed Pb is rapidly absorbed, mostly binds to red blood cells, and circulates throughout the body [26]. Therefore, it may migrate rapidly to lymphoid tissues and organs where immune cells reside and affect them. In particular, when translocated to the thymus in mice (*Mus musculus*), Pb induces suppression of immune function [27], whereas in carp (*Cyprinus carpio*), it induces oxidative stress, causing a deviation of immune function from homeostasis [28]. In humans, Pb has been reported to cause a decrease in immunoglobulins [29], although its effects on organs are rarely reported due to the lack of in vivo studies.

Considering previous studies of the immunological effects of Pb pollution, gulls with advanced Pb pollution may have a reduced ability to produce antibodies, including those specific for avian influenza viruses. Therefore, rather than the impacts of direct intake of Pb as in the California condors and white-tailed ducks, low levels of Pb pollution suppress immune function and are likely to impair their defense responses against viruses and cause mass mortalities.

The black-headed gull (*Chroicocephalus ridibundus*) is a migratory species, and influenza A infection has been reported in European countries, Asian countries such as China and Mongolia, and Russia [30,31,32,33,34,35]. The gull populations inhabiting high-latitude regions have an ecology of migration [36]. The black-headed gull is the primary species in which the H16 subtype of the influenza A virus has been reported [34]. The H16 subtype is also very rarely found in duck species [37]. Thus, this gull species plays an important role as a carrier of the influenza A virus [31]. Moreover, it is expected that gull species carry on the diversification of viral genes through various characteristics, such as patterns of replication and transmission differing from those observed in ducks [38].

Black-headed gulls tend to select areas inhabited by humans, such as urban areas and farming villages, as habitat environments. Thus, among wild bird species, black-headed gulls are susceptible to environmental pollutants, and cases of contamination have often been reported [39,40,41,42]. As a result of a Pb contamination case, a decrease in reproductive success rate has been reported [43,44,45,46].

In this study, we hypothesized that Pb pollution negatively affects avian influenza virus-specific antibody production in black-headed gulls. We aimed to investigate the seasonal changes in Pb pollution in black-headed gulls and to clarify the relationship between the Pb pollution level and the avian influenza virus-specific antibody titer.

## 2. Materials and Methods

### 2.1. Study Area

A total of 170 birds were captured in Tokyo Bay, Chiba Prefecture, Japan (35° N, 139° E) from January 2019 to April 2020 (Tokyo Bay population; TBP) and in Mikawa Bay, Aichi Prefecture, Japan (34° N, 137° E) from November 2019 to April 2021 (Mikawa Bay population; MBP) (Figure 1). In the bird-banding procedure, black-headed gulls wintering in the two regions were fitted with metal rings in the MBP and colored metal rings in the TBP [47]. This survey was conducted on days when more than half of the gulls comprising each population consisted of tagged gulls.

The birds in TBP and MBP were captured by hand, noise trap, or whoosh net. Captured birds were weighed, and blood (less than 0.5% of body weight) was collected from the basilic veins with a 1.0 mL syringe (Nipro, Osaka, Japan) attached to a heparinized 26G needle (Terumo, Tokyo, Japan). After confirming hemostasis, a plastic color ring was attached to the left tarsus in the TBP, and a metal ring of the Ministry of the Environment, Government of Japan, was attached to the TBP and MBP, and the birds were released.

### 2.2. Age and Sex

Age was determined based on plumage. Specifically, the gulls were classified into two categories: “Yearlings”, which were born in that year and had juvenile feathers, and “Adults”, aged one year or older and having no juvenile feathers [36]. Sex identification was conducted by molecular biological examination using a polymerase chain reaction (PCR). The target gene was the chromodomain helicase DNA binding protein (CHD) gene on the sex chromosome [48], and the primer used was 2550F/2718R, which is commonly used in avian species [49].

### 2.3. Pb Investigation

The blood Pb level (BLL) was considered the polluted Pb level. Pb per 1 dL of blood (µg/dL) was calculated using an inductively coupled plasma mass spectrometer (ICP-MS) [50].

### 2.4. Investigation of Antibodies to AIV

In this study, we did not conduct experimental infection using captive black-headed gulls owing to the unclear background information on AIV infection in wild individuals and from the viewpoint of animal ethics. As an alternative, evaluation of antibodies of AIV in black-headed gull plasma using enzyme-linked immunosorbent assay (ELISA) was selected. ELISA is suitable for rapid screening of populations [51]. In this study, a commercially available kit (IDEXX Influenza A ELISA Test, IDEXX, Westbrook, CT, USA) was used for the ELISA. The reaction principle of this kit is a competitive ELISA assay in which antibodies in serum and anti-AIV-horseradish peroxidase conjugate are added to an antigen-coated test plate. In a study using this kit, more than half of the chickens infected with the H11N6 subtype tested positive for about 21 days following infection [52]. In Denmark, antibodies of AIV were tested in 300 pink-footed geese (*Anser brachyrhynchus*) in the fall and spring migration seasons [53]. The results revealed that antibodies of AIV were maintained for about 340 days. However, the antibody titer in AIV peaks at seven dpi, and the antibody titer in the blood persists for several months thereafter. It is considered that the detection period of antiviral antibodies in avian serum with this kit is less than one year from 21 dpi.

In this study, an existing ELISA kit protocol was modified and used for the purpose of considering the magnitude of absorbance as a measure of antibody expression. Accordingly, the following steps were performed when using the ELISA kit. First, a neutralization test using the AIV subtype (Appendix A) was performed with sera from 16 black-headed gulls. The neutralization test was performed using the method described by Ito et al. (2021) [54], except that only replication-competent viruses were used for the antigen-antibody reaction. A positive result in the neutralization test was determined by confirming CPE through crystal violet staining of infected cells. According to the results, two gulls showed neutralizing activity against H10, which is considered a gull-origin virus subtype [55], and 14 gulls showed no neutralizing activity against all subtypes. Next, a competitive ELISA was performed using the sera from these 16 gulls according to the kit protocol. The results were inconsistent, with sera that tested negative in the neutralization test having positive results. Therefore, the ELISA kit was modified to an indirect ELISA assay without the use of the conjugate, and the goat-derived secondary antibody anti-bird IgY (H&L) (HRP; Abcam plc, Cambridge, UK) was used alternatively. After the test was performed again with a modified protocol using the same serum, the same results as those of the neutralization test were obtained. Fisher’s exact probability test showed no difference between the positive/negative results obtained with this kit and those of the neutralization test (*p* = 0.47). Considering these steps, the modified indirect ELISA protocol was used in this study.

The cutoff value for this test was the value obtained by multiplying the dispersion value of the optical density (OD) of the negative control, which was included in the kit, by three and adding the OD of the negative control. As a result, the median value of the cutoff value, mean OD + 3 SD, for this test was 1.30 OD (1.90–0.40). A judgment of negative/positive effects was performed based on this value. In addition, OD was used to assess the expression of antibodies in serum, with reference to He et al. [56]. In this study, the result measured with this kit (negative/positive) was defined as “infection history”, and the absorbance used for the result of this kit was defined as “antibody titer”.

### 2.5. Statistical Analysis

As described in this paper, we performed a retrospective cohort study of the interaction between BLL and antibody titer with reference to [57,58]. Before the analysis, we confirmed that there was no significant difference in BLL and the infection history year-to-year for each population in each of the two periods. Based on the aggregated results of the bird-banding investigation, the period during which the number of gulls increased continuously was defined as “Arriving”. and the period in which no continuous increase was confirmed was defined as “Wintering”. During the arrival period, BLL (W = 1145.44, *p* < 0.01) and infection history (Pearson chi-square = 7.61, *p* < 0.01) were significantly higher in TBP. During wintering, infection history was significantly higher in MBP than TBP (Pearson chi-square = 10.85, *p* < 0.01), but there was no significant difference in Pb (*W* = 38,550.23, *p* = 0.57). Therefore, statistical analysis was performed for each population. BLL and the infection history between the two populations were compared using the Wilcoxon two-sample and Pearson chi-square tests with Fisher’s exact *p* values.

First, to confirm seasonal changes in BLL and infection history, we counted the number of gulls in each study area. We then compared the median value of BLL and infection history in each population between the two time periods using Wilcoxon two-sample tests. Several biotic factors have been known to influence infection history. For example, according to previous studies, weak physical conditions [59] and younger ages [60] are considered to reduce antibody production. In addition, sex hormones act differently on immune cells in each sex [61,62]; thus, there may be sex-related differences in antibody production.

Therefore, the Shapiro–Wilk test was performed on the antibody titer in the gulls, which had an infection history. The antibody titer was log-transformed when the data was not normally distributed. A linear regression analysis was performed using body weight, age, and sex as explanatory variables and the antibody titer as the objective variable, and whether there was a causal relationship was tested by calculating the 95% confidence interval (95% CI), coefficient (Coef.), and *p* value. Finally, we tested for a causal relationship between the antibody titer as the objective variable and the BLL as the explanatory variable. When there was a difference in BLL between periods, the period when the sample size was larger was analyzed. A significance level of 0.05 was used in all analyses. Stata (ver. 14.0, Stata Corp., College Station, TX, USA) was used for statistical analysis.

## 3. Results

Details of the 170 captured gulls represented by population, sex, and age are shown in Table 1 and Figure 2. All data of gulls was shown in Appendix A.

The results of BLL, the infection history of AI, and the infection titer are shown in Table 2. A total of 103 gulls had an infection history. There were no significant year-to-year differences in BLL (MBP, *W* = 1224.63, *p* = 0.34; TBP, *W* = 9503.75, *p* = 0.05) and infection history (MBP, Pearson chi-square = 0.72, *p* = 0.39; TBP, Pearson chi-square = 0.00, *p* = 1.00) in each population. In contrast, infection history was significantly different in TBP (Pearson chi-square = 4.47, *p* = 0.04, positivity rate in the arriving period was 33.3% and in the wintering period was 55.7%), but not in MBP (Pearson chi-square = 0.08, *p* = 0.77, infection history in the arriving period was 85.4% and in the wintering period was 81.8%) (Table 3).

A comparison of the median BLL between arriving and wintering periods in the two populations showed no significant change in the MBP (*W* = 126.13, *p* = 0.79) but a significant increase in the TBP (*W* = 2363.50, *p* < 0.01).

The error value between the plates used in the ELISA test was 6.4% (error range: 1.7–7.9%). A linear regression was performed using antibody titer, considering the error value and body weight, age, and sex. There was no significant causal relationship between the antibody titer and body weight (MBP, 95% CI = −0.00–0.00, *p* = 0.76; TBP, 95% CI = −0.00–0.00, *p* = 0.49), age (MBP, 95% CI = −0.27–0.12, *p* = 0.43; TBP, 95% CI = −0.11–0.51, *p* = 0.21), or sex (MBP, 95% CI = −0.23–0.06, *p* = 0.23; TBP, 95% CI = −0.05–0.17, *p* = 0.28) in this study population. An analysis of BLL and antibody titer suggested that antibody titer was significantly decreased owing to increased BLL in MBP (*n* = 44, 95% CI, −0.02–0.00, *p* = 0.04, Coef. = −0.01, Figure 3). Considering the difference in Pb pollution by period, TBP was analyzed only during the wintering, and the antibody titer of TBP was also significantly decreased owing to increased BLL (*n* = 49, 95% CI, −0.02–0.00, *p* = 0.03, Coef. = −0.12, Figure 3).

## 4. Discussion

The BLL of the gulls in this study was generally below the avian standard polluted values (20 μg/dL), which were proposed in the cases of Pb pollution in raptors and waterfowl [16]. Specifically, Franson and Pain [63] indicated 0.2–0.5 ppm Pb (wet weight) as a general subclinical value of avian standard pollution [64]. The results of this study suggest that even levels below the standard polluted values observed in raptors and waterfowl, as well as levels similar to the subclinical values, show a significant relationship with antibody titer in gull populations.

This study did not target the antibody titer using a neutralization test; thus, there are few previous reports that can be directly compared with the results of this study. Teitelbaum et al. [65] investigated the relationship between mercury levels and the serotiter of antibodies to AIV in eleven wild duck species in California. They reported that increased mercury pollution significantly increased influenza A titer.

As a limitation, this study did not select the method using antibody titers obtained in neutralization tests, and therefore, we could not compare the findings with those of previous studies that included research on antibody titers using neutralization tests. Age was not significant owing to the small sample size in this study. Considering the results derived from this study, the differences in BLL in the arriving and wintering periods between populations were considered to be due to differences in sample size and regional characteristics.

In this study, BLL and infection history varied between seasons in MBP and TBP. In MBP, neither BLL nor infection history changed between the arriving and wintering periods, whereas in TBP, both BLL and infection history increased significantly from the arriving period to the wintering period. As of March 2022, both Mikawa Bay and Tokyo Bay had 24.36 to 29.52 ppm of lead deposited in the bottom sediment [66]. In order to elucidate the regional differences shown in this survey, it is necessary to consider the Pb pollution in the organisms that black-headed gulls feed on [67]. In terms of infection history, the S/N ratio of influenza A antibody was maintained almost constant for three weeks to half a year after infection in the infection experiment using European starlings (*Sturnus vulgaris*) [68]. It is reported that the half-life of Pb in the blood is approximately two weeks [69], but after it is absorbed into the body, Pb travels through the bloodstream and migrates to the lymphatic tissue, quickly affecting the expression of immune cells and molecules [70]. Considering these findings, it is possible that the TBP was infected with influenza A virus and was contaminated with Pb in Japan, and that this had some effect on the expression of influenza A virus antibodies. On the other hand, in the MBP, there was no seasonal difference in Pb concentration or infection history. This result suggests that the relationship between Pb pollution and Influenza A antibody expression can be considered irrespective of the period in MBP.

In the present study, interestingly, the results suggested that antibody production decreased in individuals with higher Pb pollution levels. The relationship between Pb pollution and antibody production can be explored in studies based on Pb exposure experiments on human immune cells, mice, and rats (*Rattus rattus*) [71,72,73,74,75]. Pb has different effects on the molecules and cells of the immune system. For example, Pb suppresses the functions of IgM-type B cells, helper T cells, TNFα, IFN-γ, and IL-2, which contribute to defense against viral infections [71,72,74,75], but enhances the functions of IL-1 and 8, which induce inflammatory responses [73]. Considering these previous studies on the effects that the Pb pollution-induced inflammatory response has on the body, Pb pollution potentially induces susceptibility to infection by suppressing TNFα, IFN-γ, and IL-2 and eliciting an inflammatory response to facilitate influenza A virus infection specifically. In addition to Pb, such effects on immune functions have been reported for heavy metals, and the common mechanisms are the enhancement of inflammatory reactions and the induction of allergic symptoms due to suppression of autoimmune upregulation [76]. Among them, Pb suppresses the production of IL-10, which suppresses immune responses, and weakens the function of macrophages involving the presentation of antigen to antibody-producing cells [76,77,78]. This effect is considered to suppress antibody production and can be considered one of the factors that caused the relationship between antibody expression and Pb pollution shown in the results of this study.

Results based on the black-headed gulls in the TBP suggest that gulls can possibly be contaminated with Pb in Japan. If Pb affects antibody production of AIV, there are six concerns for Pb-polluted birds: (1) increased risk of AIV infection; (2) increased risk of onset; (3) increased risk of disease severity; (4) delayed recovery period of AI; (5) increased viral shedding load; and (6) extension of the viral shedding period. Concerns (1) to (4) are considered to cause health damage at the individual level, and (5) and (6) are considered to be factors that promote viral contamination of other birds of the same or different species, including humans and the environment. Pb is still widely used in human society and is resultingly discharged into the environment [79]. Ishii et al. [80], who examined the stable isotopes of Pb found in wild Steller’s sea eagles (*Haliaeetus pelagicus*) and white-tailed sea eagles (*Haliaeetus albicilla*), detected isotopes derived from artificial Pb-based products, such as Pb ammunition. These results suggest that Pb discharged from human society is contaminating wild birds. Furthermore, AIV is reportedly transmitted from wild birds to both poultry [81] and zoo animals [82]. In particular, highly pathogenic AIV H5N1 subtypes have been previously detected in black-headed gulls [83]. Combining these existing reports with the results of this survey, we can speculate that environmental pollutants discharged from human society result in the spread of zoonotic diseases from wild birds to human society (Figure 4). Especially, in the case of the black-headed gull, it has a migratory strategy of flying short distances while taking breaks, unlike the Anatidae group [84]. Although rigorous verification is required, the study findings suggest that contamination of wild birds with Pb in the environment may facilitate the spread of zoonotic diseases and that environmental pollutants can threaten human health. Therefore, from the perspective of One Health [85], it is important to take measures to prevent further excessive Pb emissions.

## 5. Conclusions

The findings of this study indicated that lead (Pb) pollution has a negative effect on the antibody production of avian influenza viruses (AIV). This study suggests a relationship between the environmental pollutant Pb and AIV infection and considers important findings that connect infectious diseases with wild bird ecology. Wild birds, especially spring migration species, are more likely to carry AIV to stopping and breeding areas if they are contaminated with Pb.

## Figures and Tables

**Figure 1 animals-13-02338-f001:**
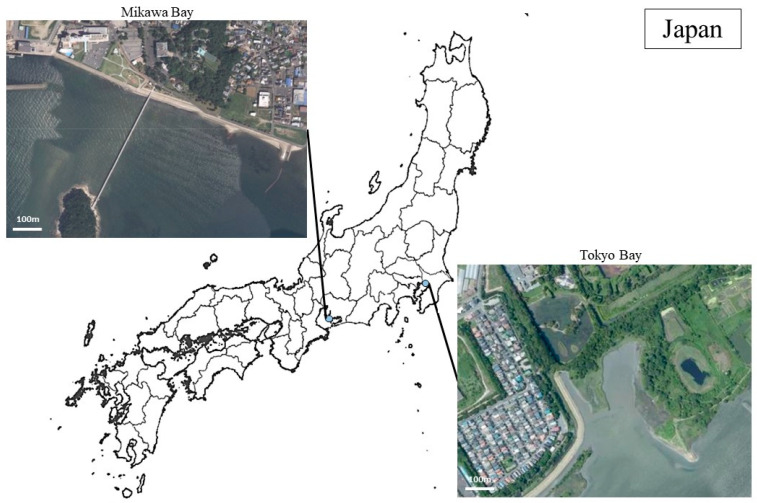
Study area. Images were cited from Google (n,d). [Google Earth satellite images of Mikawa Bay and Tokyo Bay]. Accessed on 29 May 2023, from https://www.google.com/maps.

**Figure 2 animals-13-02338-f002:**
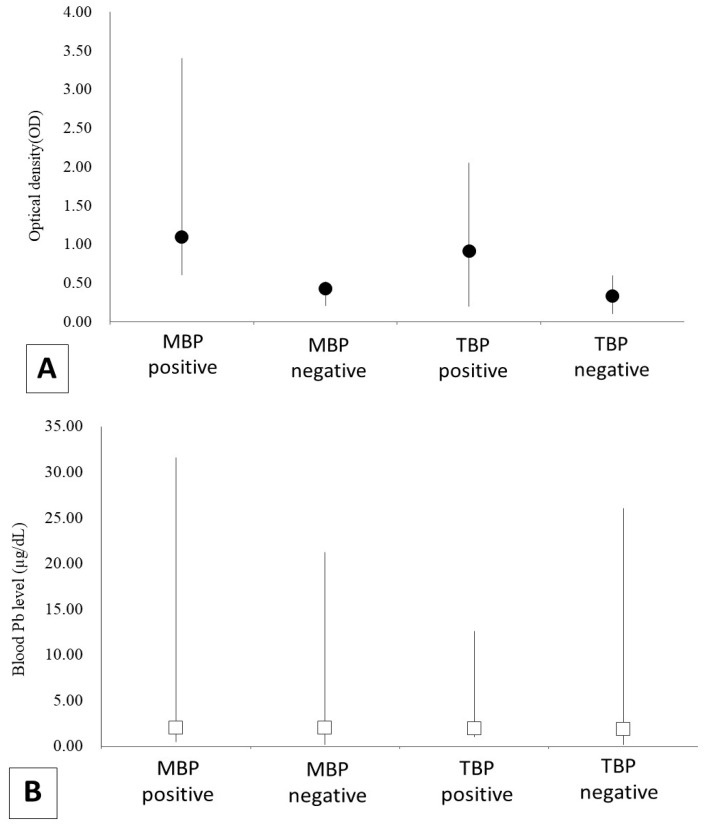
The results of the analysis of blood Pb level (BLL) and antibodies of the avian influenza virus (AIV) in 170 black-headed gulls (*Chroicocephalus ridibundus*) that were captured in Tokyo Bay from November 2018 to April 2021 (Tokyo Bay population; TBP) and in Mikawa Bay from January 2019 to April 2021 (Mikawa Bay population; MBP). (**A**) shows the result of antibody titer, and (**B**) shows BLL. Optical density (OD) was indicated black dots and BLL was white boxes.

**Figure 3 animals-13-02338-f003:**
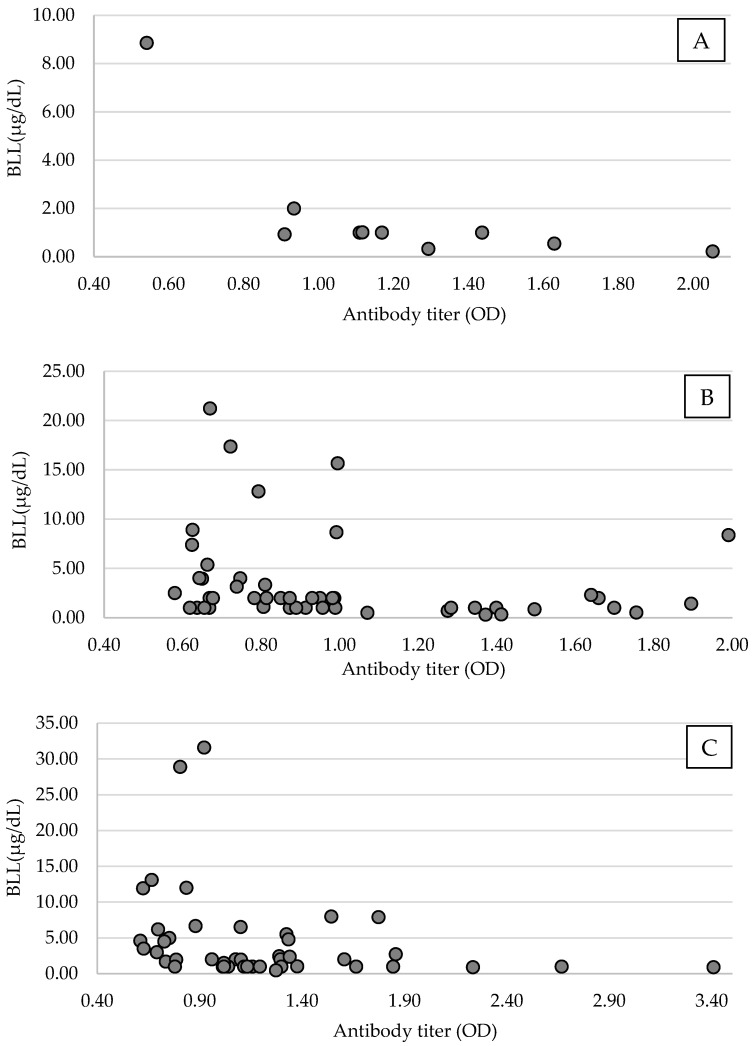
Scatter plots of blood Pb level (BLL) and antibodies of the avian influenza virus (AIV) of antibody-positive Tokyo Bay population (TMB) black-headed gulls (*Chroicocephalus ridibundus*) in the arriving period ((**A**), *n* = 10), in the wintering period ((**B**), *n* = 49), and Mikawa Bay population (MBP; (**C**), *n* = 44) from November 2018 to April 2021 in Tokyo Bay and from January 2019 to April 2021 in Mikawa Bay. Among the antibody-positive groups, antibody expression tended to decrease with higher BLL.

**Figure 4 animals-13-02338-f004:**
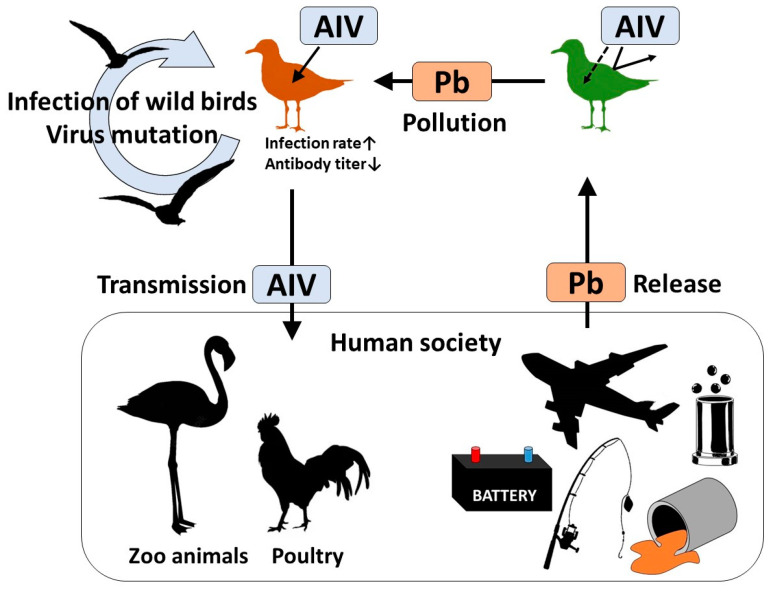
Schematic flow diagram of Pb pollution and AIV infection based on the results of this survey. Pb is released into the environment by human society [79]. Contamination associated with man-made lead products has been reported in wildlife [80]. AIV is thought to be transmitted from wild birds to both poultry and zoo animals managed by human communities [81,82]. Occasionally, AIV infections spread among wild birds, resulting in mass mortality [86]. Combining these previous reports with the results of this survey that the AIV antibody titer was significantly affected by BLL, it can be suggested that the Pb released from human activity is indirectly returned to society in the form of transmissible pathogens.

**Table 1 animals-13-02338-t001:** Details of 170 black-headed gulls (*Chroicocephalus ridibundus*) that were captured in Tokyo Bay from November 2018 to April 2021 (Tokyo Bay population; TBP) and in Mikawa Bay from January 2019 to April 2021 (Mikawa Bay population; MBP).

Periods		Arriving		Wintering	
Age		Adults	Yearlings	Adults	Yearlings
TBP	Male	17	0	42	0
	Female	13	0	42	4
MBP	Male	3	0	8	6
	Female	7	1	27	0

**Table 2 animals-13-02338-t002:** The results of the analysis of blood Pb level (BLL) and antibodies of the avian influenza virus (AIV) in 170 black-headed gulls (*Chroicocephalus ridibundus*) that were captured in Tokyo Bay from November 2018 to April 2021 (Tokyo Bay population; TBP) and in Mikawa Bay from January 2019 to April 2021 (Mikawa Bay population; MBP).

Population		MBP	TBP
Sample number (*n*)		52	118
Sex (*n*)	Male	17	59
	Female	35	59
Age (*n*)	Yearlings	7	4
	Adults	45	114
Positive rate (positive number/total number; %)		84.6 (44/52)	50.0 (59/118)
Median value of antibody titer in the positive population (OD)		1.10	0.93
Maximum and minimum values of antibody titer in the positive population (OD)		3.41–0.61	2.05–0.39
Median value of antibody titer in the negative population (OD)		0.43	0.33
Maximum and minimum values of antibody titer in the negative population (OD)		0.52–0.21	0.60–0.10
Median value of BLL in the positive population (µg/dL)		2.00	2.00
Maximum and minimum values of BLL in the positive population (µg/dL)		31.60–0.48	21.23–0.22
Median value of BLL in the negative population (µg/dL)		1.95	1.87
Maximum and minimum values of BLL in the negative population (µg/dL)		12.62–1.00	26.05–0.18

**Table 3 animals-13-02338-t003:** The results of the analysis comparing the gulls in the arriving and wintering periods based on the blood Pb level (BLL) and antibodies of the avian influenza virus (AIV) in 170 black-headed gulls (*Chroicocephalus ridibundus*). The gulls were captured in Tokyo Bay from November 2018 to April 2021 (Tokyo Bay population; TBP) and in Mikawa Bay from January 2019 to April 2021 (Mikawa Bay population; MBP).

Populations	Periods (*n*)	Items	BW (g)	BLL (µg/dL)	Antibody Titer (OD Value)	Infection History (Positive Rate, %)
Tokyo Bay	Arriving (30)	Median value	276.0	0.67	0.32	33.3%
		Max to min value	301.0–227.0	8.86–0.18	2.05–0.10	
	Wintering (88)	Median value	288.5	2.00	0.62	55.7%
		Max to min value	365.5–190.0	26.1–0.3	1.89–0.12	
Mikawa Bay	Arriving (11)	Median value	274.5	1.30	0.92	81.8%
		Max to min value	317.0–214.6	31.60–0.48	2.67–0.43	
	Wintering (41)	Median value	278.5	2.00	1.00	85.4%
		Max to min value	358.0–222.1	28.90–0.91	3.41–0.21	

## Data Availability

The data presented in this study are available from the corresponding author upon reasonable request.

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
