# Peer review of "Evaluation of the Effect of Pb Pollution on Avian Influenza Virus-Specific Antibody Production in Black-Headed Gulls (Chroicocephalus ridibundus)"

_animals, 2023, doi:10.3390/ani13142338_

Round 1
Reviewer 1 Report
The design of the experiment lacks of the logic of science. The authors selected total 170 gulls in the duration of four years, however, the birds are not the same birds in different time, so there is no quarrel relevance for the level of BLL and antibody against avian influenza viruses in different time.
The authors stated that there were six concerns about the Pb-polution for birds in the Abstract. However, to the readers, these concerns are likely regarded as hypothesis in the part of Discussion, because there was not any data in the manuscript supporting these ideas.
The quality of English is fine for me.
Author Response
Respond to Reviewer 1 Comments
The design of the experiment lacks of the logic of science. The authors selected total 170 gulls in the duration of four years, however, the birds are not the same birds in different time, so there is no quarrel relevance for the level of BLL and antibody against avian influenza viruses in different time.
The authors stated that there were six concerns about the Pb-polution for birds in the Abstract. However, to the readers, these concerns are likely regarded as hypothesis in the part of Discussion, because there was not any data in the manuscript supporting these ideas.
>The change over time of antibody expression, the duration of Pb effect on immune cells, and the half-life of Pb in blood are shown in the discussion, and the validity of the cross-sectional evaluation is explained in the discussion.
L430-441: In terms of infection history, the S/N ratio of influenza A antibody was maintained almost constant for three weeks to half a year after infection in the infection experiment using European starlings (Sturnus vulgaris) [68]. It is reported that the half-life of Pb in the blood is approximately two weeks [69], but after it is absorbed into the body, Pb travels through the bloodstream and migrates to the lymphatic tissue, quickly affecting the expression of immune cells and molecules [70]. Considering these findings, it is possible that the TBP was infected with influenza A virus and was contaminated with Pb in Japan, and that this had some effect on the expression of influenza A virus antibodies. On the other hand, in the MBP, there was no seasonal difference in Pb concentration, and the infection history. This result suggests that the relationship between Pb pollution and Influenza A antibody expression can be considered irrespective of the period in MBP.
>In addition, we have revised the part of the summary that you pointed out.
L23-28: Pb pollution was evaluated based on blood lead levels (BLL) and antibody positive rate (infection history), and antibody titer was evaluated using serum. The results indicated that the antibody titer was significantly decreased owing to increased BLL in MBP. There were no significant year-to-year differences in BLL and antibody titer. In the wintering period, antibody titer was also significantly decreased owing to increased BLL in TBP. The findings of this study indicated that Pb pollution had a possible negative effect on the antibody production of AIV.

Reviewer 2 Report
Dear authors,
The manuscript "Evaluation of the Effect of Pb Pollution ... (Chroicocephalus ridibundus)" is about a very important and current subject.
However, I have some questions regarding the methodology and the results.
1) Authors must adequately justify the changes made for the use of the IDEXX Ifluenza A ELISA test kit.
The system was validated as a competitive ELISA system and not as an indirect ELISA (the authors incorrectly call it a direct ELISA), as it is possible to have confidence in the results obtained with this alteration.
2) Authors must justify the classification of positive animals with OD <1.30.
2) The authors must describe the methodology used in the neutralization test.
3) Authors must justify the use of the H10 antigen in the neutralization test.
4) The authors must describe whether the handling of the animals followed the legal guidelines.
5) The authors must describe in a figure or table the results comparing the animals that arriving and wintering.
6) Authors should describe body weight data.
Author Response
Response to Reviewer 2 comments
1) Authors must adequately justify the changes made for the use of the IDEXX Ifluenza A ELISA test kit.
The system was validated as a competitive ELISA system and not as an indirect ELISA (the authors incorrectly call it a direct ELISA), as it is possible to have confidence in the results obtained with this alteration.
>In this study, we wanted to use the absorbance (S/N ratio) as a measure of antibody expression, so we modified the existing ELISA kit protocol. We added a sentence with this meaning and corrected the terminology from direct to indirect.
L179-180: In this study, an existing ELISA-kit protocol was modified and used for the purpose of considering the magnitude of absorbance as a measure of antibody expression.
L191-198: Therefore, the ELISA kit was modified to an indirect ELISA assay without the use of the conjugate and the goat-derived secondary antibody anti-bird IgY H&L (HRP; Abcam plc, Cambridge, UK) was used alternatively. After the test was performed again with a modified protocol using the same serum, the same results as that of the neutralization test were obtained. Fisher’s exact probability test showed no difference between the positive/negative results obtained with this kit and that of the neutralization test (p = 0.47). Considering these steps, the modified indirect ELISA protocol was used in this study.
2) Authors must justify the classification of positive animals with OD <1.30.
>As the cutoff value, "mean OD + 3SD", which is commonly used for positive/negative determination of ELISA, was used. Additionally, plates have different cut-off values, we have indicated the median value, as well as the maximum and minimum cut-off values.
L201-202: As a result, the median value of cutoff value, mean OD + 3SD, for this test was 1.30 OD (0.38–1.90). A judgment of negative/positive effects was performed based on this value.
3) The authors must describe the methodology used in the neutralization test.
>A table of the viruses used has been added as Supplementary Table 1, and the neutralization test method also added in the paragraph of Investigation of antibodies of AIV.
L181-186: First, a neutralization test using AIV subtype (Supplementary Table 1) was performed with sera from 16 black-headed gulls. The neutralization test was performed using the method by Ito et al. (2021) [54], except only replication-competent viruses were used for antigen-antibody reaction. A positive result in the neutralization test was determined by confirming CPE through crystal violet staining of infected cells.
4) Authors must justify the use of the H10 antigen in the neutralization test.
> The H10 subtype was reportedly derived from seagulls, so it may have been easier to detect than other subtypes. In this manuscript, we added this consideration as a brief explanation sentence.
L186-188: According to the results, two gulls showed neutralizing activity against H10 which is considered to gull-origin viruses’ subtype [55],
5) The authors must describe whether the handling of the animals followed the legal guidelines.
>A permit from the municipality is related to the Wildlife Conservation Act. So, we added this content in Institutional Review Board Statement.
L534-538: Institutional Review Board Statement: The field study was approved by the Experimental Animal Ethics Committee of Nippon Veterinary and Life Science University (approval number: 30s–47, 2020K–78), Chiba Prefecture (Chiba Prefectural Directives Nos. 1568, 1667, and 900) and Aichi Prefecture (31 Tokan No. 588–5, 2 Tokan No. 1013–3). These municipal permits are in accordance with the Wildlife Protection, Control, and Hunting Management Act of the Ministry of the Environment.
6) The authors must describe in a figure or table the results comparing the animals that arriving and wintering.
7) Authors should describe body weight data.
>We added Table 3 as a response for your comments.
L335-339: Table 3. The results of the analysis comparing the gulls in the arriving and the wintering periods based on the blood Pb level (BLL) and antibodies of the avian influenza virus (AIV) in 170 black-headed gulls (Chroicocephalus ridibundus). The gulls were captured in Tokyo Bay from No-vember 2018 to April 2021 (Tokyo Bay population; TBP) and in Mikawa Bay from January 2019 to April 2021 (Mikawa Bay population; MBP).

Reviewer 3 Report
Evaluation of the Effect of Pb Pollution on Avian Influenza 2 Virus-Specific Antibody Production in Black-Headed Gulls 3 (Chroicocephalus ridibundus)
Ushine et al
1-Summary
The article by Ushine et al attempts to investigate the link between lead (Pb) contamination and vulnerability to avian flu in a gull species. In particular, the authors highlight the fact that lead contamination interferes with the immunocompetence in many species: consequently, common gulls contaminated with lead should be more prone to infection. Since the black headed gull lives in polluted environments, it is more prone to be contaminated with lead and thus more vulnerable to infection.
The topic is interesting and deserves attention, since the secondary effects of metal contamination are little studied.
2-General concept comments.
Although the objective of the study is interesting and addresses an important topic, the research approach shows some weaknesses that need to be clarified. In the following lines, I will list all the unclear points and doubts that the authors need to clarify:
1) The authors explain that they have divided the birds into two groups based on the stability of the number of the birds: 'arriving' and 'wintering'. However, this method is highly questionable: the fact that a bird is caught in the 'arriving' phase does not mean that it is a migrant. It could be an individual that has been present for some time and is preparing to overwinter in the area. In my opinion, the only ways of knowing whether a gull is an actual migrant are a) to have individually tagged animals, b) or a population so small that researchers can recognize the individuals joining it with a good approximation. In any case, this aspect must be clarified very precisely because it is crucial to the overall meaning of the article.
2) The basis of the authors' reasoning is that there must be a relationship between lead and flu antibodies. I am not a virologist, but I wonder why the same relationship should not apply to cadmium, mercury or to the amount of carotenoids in the diet. Again, the authors need to explain in detail the reasons linking immune response and lead concentration, or the experiments put in place to identify with certainty the effect of lead (and not that of other compounds or metals).
3) The concentration of metals in blood reflects a short term effect, while that in feathers reflects a medium or long term effect. The authors should bear this in mind when describing the cause-effect relationship between pollution and vulnerability to infection.
4) In general I find the text too assertive. I would suggest a more careful approach by the authors, expecially in the discussion and in the conclusion.
3-Specific comments
r 36-41: statistics should not be reported in Abstract
r 196-201: unclear: are you writing that you have checked the effect of (1) time and (2) population on BLL ?
r 202-206: as already annoted, this division seems rather arbitrary to me. In the “arriving” population you can have sedentary individuals and migratory individuals can arrive in the “wintering” population without you realising it, if some individuals die or disperse out of lagoons and sea.
Figure 2 B: it would be interesting to compare the 4 point. Please enlarge the scale.
r 254-258: you are stating that there is a relation between lead concentration and antibody titer. This is the focal point, the “bulk” of your paper. It is important to have a graph showing the significant effect of BLL on antibody.
r 325-327: I suggest to be less assertive. You have not demonstrated the effect of Pb contamination on immunocompetence. You are suggesting that can be a relation between the two phenomena.
Literature
I suggest Boskabady et al 2018 Environment International 404-418 and Shefa and Hérouxas 2017 Toxicology Letters, 232-237 as a good references.
I suggest an extensive revision.
Author Response
Response to Reviewer 3 comments
Major Comments
1) The authors explain that they have divided the birds into two groups based on the stability of the number of the birds: 'arriving' and 'wintering'. However, this method is highly questionable: the fact that a bird is caught in the 'arriving' phase does not mean that it is a migrant. It could be an individual that has been present for some time and is preparing to overwinter in the area. In my opinion, the only ways of knowing whether a gull is an actual migrant are a) to have individually tagged animals, b) or a population so small that researchers can recognize the individuals joining it with a good approximation. In any case, this aspect must be clarified very precisely because it is crucial to the overall meaning of the article.
> The domestic migration of black-headed gulls has been grasped since 1993 by coloring and metal ring in bird banding, and presumed their major migration routes. Based on this information, we have selected two populations. Although not all overwintering individuals, most of the population in Mikawa Bay have metal rings on their tarsus, and those in Tokyo Bay have coloring and metal rings. It is possible to identify individuals by individual number engraved on the metal ring. It's just as you said, the composition of gulls in each population is possible to be changed. Thus, we observe leg rings and confirm the composition of the population prior to the capture survey. The survey was conducted on days when it was confirmed that more than half of the entire population consisted of tagged individuals or individuals accustomed to the area (e.g., approaching the investigator). This information has been added in Materials and Methods.
L115-118: In the bird-banding procedure, black-headed gulls wintering in the two regions were fitted with metal rings in the MBP, and colored and metal rings in the TBP [47]. This survey was conducted on days when more than half of the gulls comprising each population consisted of tagged gulls.
L211-214: Based on the aggregated results of the bird-banding investigation, the period during which the number of gulls increased continuously was defined as “Arriving,” and the period in which no continuous increase was confirmed was defined as “Wintering.”
2) The basis of the authors' reasoning is that there must be a relationship between lead and flu antibodies. I am not a virologist, but I wonder why the same relationship should not apply to cadmium, mercury or to the amount of carotenoids in the diet. Again, the authors need to explain in detail the reasons linking immune response and lead concentration, or the experiments put in place to identify with certainty the effect of lead (and not that of other compounds or metals).
> After describing the effects of heavy metals on immune function, the characteristic actions of Pb were discussed.
L452-460: In addition to Pb, such effects on the immune functions have been reported for heavy metals, and the common mechanisms are the enhancement of inflammatory reactions and the induction of allergic symptoms due to suppression of autoimmune upregulation [76]. Among them, Pb suppresses the production of IL-10, which suppresses immune responses, and weakens the function of macrophages involving the presentation of antigen to antibody-producing cells [76–78]. This effect is considered to suppress antibody production, and can be considered to be one of the factors that caused the relationship between antibody expression and Pb pollution shown in the results of this study.
3) The concentration of metals in blood reflects a short term effect, while that in feathers reflects a medium or long term effect. The authors should bear this in mind when describing the cause-effect relationship between pollution and vulnerability to infection.
> The study used only blood for the purpose of measuring lead levels. If my understanding is insufficient, please give me additional advice.
4) In general I find the text too assertive. I would suggest a more careful approach by the authors, expecially in the discussion and in the conclusion.
> We have revised the assertion in the discussion and conclusion to mean that it is one of the possibilities.
Discussion
L458-460: This effect is considered to suppress antibody production, and can be considered to be one of the factors that caused the relationship between antibody expression and Pb pollution shown in the results of this study.
L508-509: Figure 3. Schematic flow diagram of Pb pollution and AIV infection based on the results of this survey.
Conclusion
L519-523: This study suggests a relationship between the environmental pollutant Pb and AIV infection, and considers important findings that connect infectious diseases with wild bird ecology. Wild birds, especially spring migration species, are more likely to carry AIV to stopping and breeding areas if they are contaminated with Pb.
3-Specific comments
r 36-41: statistics should not be reported in Abstract
> Based on your advice, I removed the statistics in the abstract.
L37-41: A significant increase was found in the TBP. A decrease in BLL significantly increased antibody titer during wintering in TBP and MBP. Pb pollution had a negative effect on the production of the AIV antibodies. These findings suggest that wild birds which were contaminated by Pb in the envi-ronment may facilitate the spread of zoonotic diseases, further increasing the possibility that environmental pollutants may threaten human health.
r 196-201: unclear: are you writing that you have checked the effect of (1) time and (2) population on BLL ?
r 202-206: as already annoted, this division seems rather arbitrary to me. In the “arriving” population you can have sedentary individuals and migratory individuals can arrive in the “wintering” population without you realising it, if some individuals die or disperse out of lagoons and sea.
>Our response to this advice is the same as Major Comment 1. Additionally, we added some sentences to the Materials and Methods and Results.
Materials and Methods
L214-218: During the arriving period, BLL (W = 1145.44, p < 0.01) and infection history (Pearson chi-square= 7.61, p < 0.01) were significantly higher in TBP. During wintering, infection history was significantly higher in MBP than TBP (Pearson chi-square = 10.85, p < 0.01), but there was no significant difference in Pb (W = 38550.23, p = 0.57). Therefore, statistical analysis was performed for each population.
Results
L304-308: In contrast, infection history was significantly different in TBP (Pearson chi-square = 4.47, p = 0.04, positivity rate in the arriving period was 33.3% and in the wintering period was 55.7%), but not in MBP (Pearson chi-square = 0.08, p = 0.77, positivity rate in the arriving period was 85.4% and in the wintering period was 81.8%) (Table 3).
Figure 2 B: it would be interesting to compare the 4 point. Please enlarge the scale.
> This figure has been enlarged to the extent permitted by the regulations of your journal.
r 254-258: you are stating that there is a relation between lead concentration and antibody titer. This is the focal point, the “bulk” of your paper. It is important to have a graph showing the significant effect of BLL on antibody.
> Following your advice, we have added Figure 3.
L397-402: Figure 3. Scatter plots of blood Pb level (BLL) and antibodies of the avian influenza virus (AIV) of antibody-positive Tokyo Bay population (TMB) black-headed gulls (Chroicocephalus ridibundus) in the arriving period (A, n = 10), in the wintering period (B, n = 49), and Mikawa Bay population (MBP; C, n = 44) from November 2018 to April 2021 in Tokyo Bay and from January 2019 to April 2021 in Mikawa Bay. Among the antibody-positive groups, antibody expression tended to decrease with higher BLL.
r 325-327: I suggest to be less assertive. You have not demonstrated the effect of Pb contamination on immunocompetence. You are suggesting that can be a relation between the two phenomena.
> Our response to your advice is the same as Major Comment 4. We have only mentioned the possibility suggested by this survey result.
Literature
I suggest Boskabady et al 2018 Environment International 404-418 and Shefa and Hérouxas 2017 Toxicology Letters, 232-237 as a good references.
> We appreciate for you to introduce good quality literatures. We have read these literatures and quoted the Boskabady et al (2018) in Discussion.

Round 2
Reviewer 2 Report
Dear authors,
The changes made to the manuscript were significant and significantly improved it.
However, I have concerns about the results presented.
Figure 2A and Table 2 show the levels of antibodies in the peripheral blood of the animals. And they are divided into positive and negative animals for infection by AIV based on a cutoff value of 1.3.
But clearly, in the positive groups, there are animals with OD values lower than 1.3. How can the authors justify this?
It is important to note that the authors' hypothesis is that there is an inverse relationship between the levels of specific antibodies and Pb in animals with a positive avian influenza infection.
How is it possible to make this relationship from the data presented?
Author Response
Response to Reviewer 2 comments.
Figure 2A and Table 2 show the levels of antibodies in the peripheral blood of the animals. And they are divided into positive and negative animals for infection by AIV based on a cutoff value of 1.3.
But clearly, in the positive groups, there are animals with OD values lower than 1.3. How can the authors justify this?
It is important to note that the authors' hypothesis is that there is an inverse relationship between the levels of specific antibodies and Pb in animals with a positive avian influenza infection.
How is it possible to make this relationship from the data presented?
>After carefully checking the original data, it was confirmed that some antibody-negative individuals were mixed with positives in the calculation of the median, maximum, and minimum values in the process of creating Table 2. We reviewed each item in Table 2 and made corrections.
The OD value, which is the criterion for positive/negative judgment, differs between plates, thus, not all positive individuals satisfy 1.30 or higher. Regarding this standard, we have added the following explanation as a response to your first comment, so please check it.
L201-202: As a result, the median value of cutoff value, mean OD + 3SD, for this test was 1.30 OD (1.90–0.40).

Reviewer 3 Report
The paper has been improved following suggestions. It can be published.
The paper has been improved following suggestions.
It can be published.
Author Response
Response to Reviewer 3 comment.
Thank you for your prompt response and careful review of the manuscript.